# Infection and telomere length: A systematic review

**Louis Tunnicliffe** [ID][1]*, **Rutendo Muzambi**[2], **Jonathan W. Bartlett**[1], **Laura D. Howe**[3], **Khalid A. Basit**[1], **Kwabena Asare** [ID][1], **Georgia Gore-Langton**[1], **Kathryn E. Mansfield**[4], **Veryan Codd**[5], **Charlotte Warren-Gash**[1]

1 Faculty of Epidemiology & Population Health, London School of Hygiene & Tropical Medicine, London, United Kingdom, 2 Department of Epidemiology & Biostatistics, School of Public Health, Imperial College London, London, United Kingdom, 3 MRC Integrative Epidemiology Unit, Department of Population Health Sciences, Bristol Medical School, University of Bristol, Bristol, United Kingdom, 4 School of Health and Care Sciences, University of Lincoln, Leicester, United Kingdom, 5 Department of Cardiovascular Sciences, University of Leicester, Leicester, United Kingdom

* Louis.Tunnicliffe@lshtm.ac.uk

## Abstract

### Background

Infections may increase the risk of age-related diseases such as dementia. Accelerated immunological ageing, measurable by telomere length (TL), may be a potential mechanism. However, the relationship between different infections and TL or telomere attrition remains unclear. This systematic review synthesises existing evidence on whether infections contribute to TL or telomere attrition and highlights research gaps to inform future studies.

### Objective

To summarise the literature on associations between infections and telomere length or attrition.

### Methods

We conducted comprehensive searches across six databases (MEDLINE, EMBASE, Web of Science, Scopus, Global Health, Cochrane Library) from inception to 22 May 2025, using concepts of infections, TL, and study type. Two researchers independently screened studies, extracted data, and assessed risk of bias (ROB) using the ROBINS-E tool. Meta-analysis was unfeasible due to heterogeneity, so a narrative synthesis was conducted. Studies were grouped by infection type, telomere measurement assay, cell type, and statistical approach. A GRADE assessment was performed to evaluate evidence quality.

**Data availability statement:** All relevant data are within the manuscript and its Supporting Information files.

**Funding:** This systematic review was supported by Charlotte Warren-Gash's Wellcome Career Development Award [225868/Z/22/Z].

**Competing interests:** The authors have declared that no competing interests exist.

**Abbreviations: AD**, Alzheimer's Disease; **BMJ**, British Medical Journal; **C. pneumoniae**, *Chlamydia pneumoniae*; **CMV**, Cytomegalovirus; **COVID-19**, Coronavirus Disease 2019; **DNAm-TL**, DNA Methylation-Based Estimator of Telomere Length; **EBV**, Epstein-Barr Virus; **E-O**, Exposure-outcome; **FISH**, Fluorescence *In Situ* Hybridisation; **Flow-FISH**, Flow Fluorescence *In Situ* Hybridisation; **HAART**, Highly Active Antiretroviral Therapy; **HBV**, Hepatitis B Virus; **HCV**, Hepatitis C Virus; **HHV-6**, Human Herpesvirus 6; **HIV**, Human Immunodeficiency Virus; **HPV**, Human Papillomavirus; **HR-HPV**, High-Risk Human Papillomavirus; **HSV-1**, Herpes Simplex Virus Type 1; **HSV-2**, Herpes Simplex Virus Type 2; **HTLV-1**, Human T-Lymphotropic Virus Type 1; **HTLV-2**, Human T-Lymphotropic Virus Type 2; **ICD-10**, International Classification of Diseases, 10th Revision; **MDR-TB**, Multidrug-Resistant Tuberculosis; **PCR**, Polymerase Chain Reaction; **PECOS**, Population, Exposure, Comparator, Outcomes, and Study Characteristics; **PRISMA-P**, Preferred Reporting Items for Systematic Reviews and Meta-Analyses of Protocols; **PROSPERO**, International Prospective Register of Systematic Reviews; **Q-FISH**, Quantitative Fluorescence *In Situ* Hybridisation; **Q-PCR**, Quantitative Polymerase Chain Reaction; **SB**, Southern Blot; **STELA**, Single Telomere Length Analysis; **TB**, Tuberculosis; **TeSLA**, Telomere Shortest Length Assay; **TL**, Telomere Length; **TRF**, Terminal Restriction Fragment

## Results

Our searches identified 10,349 studies, of which 73 met eligibility criteria. Most (59) were cross-sectional and most were published after 2000, with the earliest from 1996. Most studies were from the USA (17). HIV was the most frequently studied infection (35 studies), with 79% (excluding overlapping samples) reporting an association between HIV and reduced TL or increased telomere attrition. Findings for other infections, including herpesviruses and Human Papillomavirus were more variable. Variation in infection type, measurement assay, cell type, and statistical approach made cross-study comparisons challenging. Most studies had a high ROB, mainly due to unmeasured confounding. The GRADE assessment rated evidence quality as very low.

## Conclusions

Our review highlights a potential link between HIV and TL and telomere attrition. More robust longitudinal studies with standardised measurements and better confounder control are needed, particularly for non-HIV infections.
PROSPERO (ID:CRD42023444854)

## Background

Infections may contribute to the development of age-related diseases such as cardiovascular disease (CVD) and dementia. Systematic reviews and meta-analyses [1,2] have shown that acute infections, including influenza and COVID-19, are associated with increased CVD risk. Observational evidence shows that a broad range of viral, bacterial and other infections leading to hospital admission are associated with increased risk of major adverse cardiovascular events [3]. Similarly, some infections may be implicated in dementia risk; severe infection syndromes including sepsis and pneumonia are associated with increased long-term dementia risk in large longitudinal studies [4,5] although evidence for association with individual pathogens such as human herpesviruses is less clear [6–8].

One potential mechanism underlying the potential infection-dementia association is accelerated immunological ageing. Immunological ageing refers to the gradual decline in immune system function and can be assessed through telomere length (TL) [9]. Telomeres are repetitive nucleotide sequences at the ends of chromosomes that protect genetic material from degradation. With each cell division, telomeres progressively shorten due to incomplete end-replication. Ultimately, this leads to cellular senescence or apoptosis [10]. Studies have linked shorter telomeres to Alzheimer's disease (AD). One meta-analysis found that individuals with AD have shorter telomeres compared to those without AD [11]. Similarly, a Mendelian randomisation study found that short telomeres were associated with increased risk of AD in both observational and genetic analyses [12].

However, the association between infections and TL remains unclear. Existing studies vary in terms of the infections studied, whether infections are acute or chronic, and the assays used to measure TL [13–15]. There are also variations in the measures of TL used in existing studies as well as study type and statistical analysis method, making it difficult to aggregate evidence [15–19].

Therefore, our systematic review aimed to summarise all available research on the association between infections and TL, or attrition, in adult humans, considering various study designs, telomere measurement methods, and statistical analysis approaches.

## Methods

### Protocol and registration

Our systematic review followed the guidelines outlined in the Preferred Reporting Items for Systematic Reviews and Meta-Analyses of Protocols (PRISMA-P) statement (S1 File). We also pre-registered (PROSPERO registration number CRD42023444854) and published our protocol prior to starting the review [20–22] (S2 File).

### Eligibility criteria

Studies were considered for inclusion in the systematic review if they met the Population, Exposure, Comparator, Outcomes, and Study characteristics (PECOS) framework criteria [23] presented in Table 1.

### Information sources

We searched for published studies across six bibliographic databases: MEDLINE (Ovid interface), EMBASE (Ovid interface), Web of Science, Scopus, Global Health, and the Cochrane Library from database inception to August 31, 2023. The reference lists of included papers were manually examined to find any other relevant studies.

### Search

Our search strategy included three concepts: infections, TL, and study type. We combined search concepts using the Boolean 'AND' operator. We combined key words with database-specific subject headings for each search concept. We developed and adapted our search for various databases with guidance from a librarian at the London School of Hygiene

**Table 1. Summary of systematic review inclusion eligibility criteria.**

| Population | Humans, adults aged 18 years or older, any geographic area and any setting. |
|---|---|
| **Exposure** | Any infection (i.e., any pathogen, site, severity, acute or chronic). Infection diagnosis could have been determined through electronic healthcare records (e.g., using morbidity-coded diagnoses), self-report, antibody measurements, or other laboratory markers of infection. In Mendelian randomisation studies, the exposure was defined as genetic variants linked to infection. |
| **Comparator** | The comparison group differed depending on the type of study. Individuals who were not exposed to the infection served as the comparator group for both cross-sectional and cohort studies. In case-control studies, the comparison group consisted of individuals who had normal TL. |
| **Outcome** | The outcomes included TL and telomere attrition (change in TL). We included research using any established assay to measure TL [24]. These assays included PCR (Polymerase Chain Reaction), TRF (Terminal Restriction Fragment) analysis, STELA (Single TL Analysis), TeSLA (Telomere Shortest Length Assay) and FISH (Fluorescence In Situ Hybridisation) techniques. We did not restrict by cell type in which TL was assessed. |
| **Study Design** | To ensure all possible designs were considered, we included cross-sectional studies, case-control studies, cohort studies, randomised controlled trials of vaccines or antimicrobial treatments, and Mendelian randomisation studies. |

and Tropical Medicine. We did not restrict our search on publication date or language. The complete search strategy can be found in S3 File.

## Study selection

We initially de-duplicated the papers returned by our search using automatic and manual methods with EndNote 20 software; automatically identified duplicates were manually reviewed and verified to ensure accuracy. If two studies had study samples that completely overlapped and similar findings, only the study with the largest sample size was kept. Studies with partially overlapping samples (or the same sample if results differed between studies) were retained. For studies with completely or almost completely overlapping samples, the one with the largest sample size was kept, and if these were the same, the most recent publication was selected. Two researchers reviewed all titles and abstracts independently against the eligibility criteria. When results differed, reviewers discussed to achieve consensus regarding which articles should proceed to full text review. In cases where the two reviewers could not agree a third reviewer was consulted. We repeated the process for the full-text review.

## Data collection

The primary investigator extracted data from all included papers using a standardised data extraction form. To ensure consistency, a second researcher independently extracted data from a random sample of 10% of the included papers and compared the results with those of the primary investigator.

Our standard data extraction form captured data on study characteristics guided by the PECOS framework (S1 Table). We captured details on: 1) population characteristics (e.g., age, sex, setting): 2) infection exposure (e.g., acute or chronic, pathogen, severity); 3) comparators; 4) outcomes including TL or attrition, measurement assays (e.g., Quantitative Polymerase Chain Reaction (Q-PCR)), and cell type assessed; and 5) study characteristics (setting, study design, follow-up). The results extracted included unadjusted mean/median telomere length measurements as well as the crude and adjusted effect estimates from statistical modelling, e.g., beta coefficients from linear regression, odds ratios from logistic regression, F-values from mixed effect models. Details of any covariates and subgroup analyses were also collected (e.g., by sex or age).

## Risk of bias

Some studies reported data on multiple infections, meaning the number of exposure-outcome (E-O) relationships exceeded the number of included studies. Therefore, we assessed risk of bias for each E-O relationship individually, as different E-O relationships within the same study could have varying risk-of-bias scores.

Two researchers independently assessed risk of bias of each E-O relationship using the Risk Of Bias In Non-randomized Studies – of Exposures (ROBINS-E) tool [25]. The ROBINS-E tool evaluates risk of bias in the following domains: Domain 1-confounding, D2-measurement of the exposure, D3-selection of participants into the study (or into the analysis), D4-post-exposure interventions, D5-missing data, D6-measurement of the outcome, and D7-selection of the reported result. For each domain, assessors answered a series of questions regarding the assessed study. Scores from all domains were then synthesised into an overall risk-of-bias rating (low, some concerns, high, very high) based on the ROBINS-E algorithm and further guidance provided in the tool.

## Synthesis of results

We conducted a narrative synthesis considering the following potential sources of heterogeneity: infection type, cell type, outcome (TL or change in TL), TL measure (relative or absolute), TL measurement assay, and statistical method. Results were grouped based on these factors, starting with infection type and progressively refining by cell type, outcome,

TL measure, assay, and statistical method, with studies in the final group being comparable across all sources of heterogeneity.

We also reported the total number of E-O relationships examined, then the number of unique E-O relationships by infection type. One E-O relationship from each overlapping sample was classed as 'unique'. To determine the unique E-O relationship, priority was given to those with longitudinal designs, followed by studies with the greatest sample size and to the most recent study. We then reported the number of unique homogeneous E-O relationships (defined as homogeneous if they matched all potential sources of heterogeneity). We also presented the number of unique E-O relationships showing evidence of an association (with smaller TL or greater telomere attrition) for each infection type. Given that many relationships were based on unadjusted results, we also reported the number with matched or adjusted analyses demonstrating evidence of association.

We did not meta-analyse due to high inter-study heterogeneity, which would render pooled estimates difficult to interpret. Instead, we presented forest plots for infections with at least three studies that reported the same type of effect estimate (e.g., difference in mean telomere lengths) with 95% confidence intervals, or where both the effect estimate and its 95% confidence interval could be calculated from the available data. To avoid duplication, when two or more studies had overlapping samples, only one was presented in the forest plot, with the same prioritisation criteria used for selecting unique studies. Forest plots were generated for HIV, Cytomegalovirus (CMV), and Herpes Simplex Virus Type 1 (HSV-1), as these infections had the highest number of studies meeting the inclusion criteria.

### GRADE assessment

To assess the quality of evidence in the review we applied the GRADE (Grading of Recommendations Assessment, Development and Evaluation) approach, which provides a structured framework for rating the certainty of evidence across studies contributing to a specific outcome [26]. To ensure consistency and comparability we applied the same inclusion criteria for GRADE as for forest plots. Specifically, we included infections with at least three studies reporting the same type of effect estimate (e.g., difference in mean telomere lengths) with 95% confidence intervals, or where both the effect estimate and its confidence interval could be calculated from the available data. To avoid duplication, studies with overlapping samples were excluded using the same prioritisation strategy. This approach was taken because GRADE is intended to assess the certainty of a coherent body of evidence and applying it to highly heterogeneous studies with differing outcomes and effect types would not yield meaningful or interpretable assessments. GRADE evaluations were therefore conducted for HIV, CMV, and HSV-1, which had the highest number of sufficiently comparable studies.

The following domains were assessed using GRADE: Risk of bias, inconsistency, indirectness, imprecision, and publication bias. We rated the strength of evidence as high, moderate, low, or very low. The criteria used for determining the quality of evidence can be found in S4 File.

## Results

### Study characteristics

Our initial search identified 8,670 records, 4,987 remained after removing duplicates. Of 4,987 titles/abstracts screened, 233 were taken to full-text review. After full-text review and citation searching, 62 studies [13,14,19, 27–82] [15–17] and 85 E-O relationships were eligible for inclusion in our review. An updated search conducted on 22.06.2024 identified an additional 1,679 records, of which 11 [83–93] met the inclusion criteria, resulting in a final total of 73 studies and 105 Exposure–Outcome relationships. Fig 1 shows an adapted [94] PRISMA flow diagram summarising the results of the two searches..

Studies included were set across a range of geographical settings, with most (i.e., n = 17, 23%) from the US, followed by 7 (10%) from Canada, and the remaining from a variety of, largely high-income, countries. Nine studies presented in conference abstracts did not clearly specify their setting.

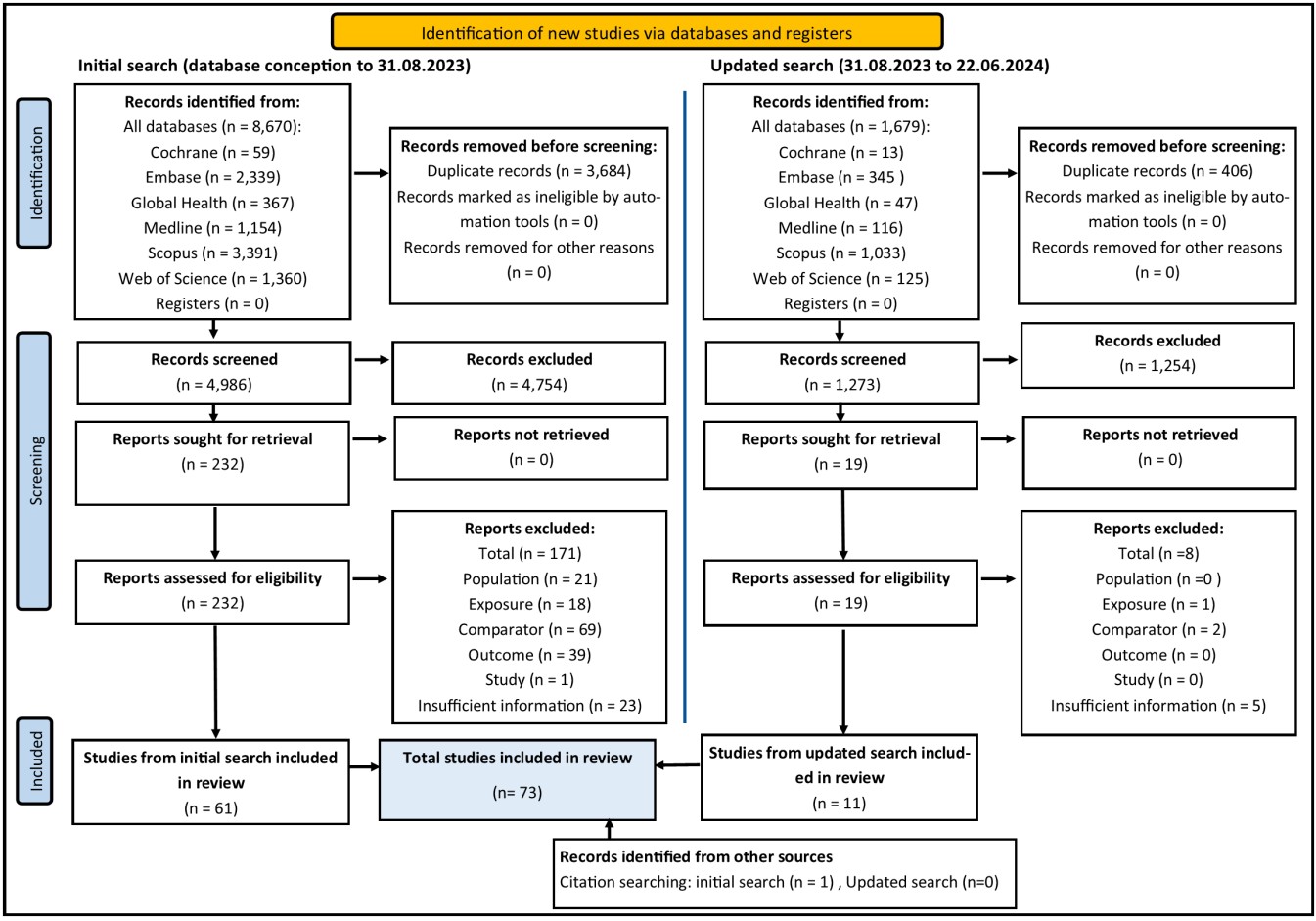

**Fig 1. PRISMA flow diagram showing how studies were selected for the systematic review.** NB- The 'Records screened' stage refers to title and abstract screening and the 'Reports assessed for eligibility' stage relates to full-text screening. Figure adapted from Page et al. (2021), PRISMA 2020 flow diagram [94].

Of the sixty-three studies included, the majority (59, 81%) were cross-sectional. Other study types included cohort, mendelian randomisation, and studies using a mixture of cross-sectional and cohort analyses. The characteristics of the included studies are presented in Table 2.

### Risk of bias assessment

Of the 105 infection-telomere relationships in the review, 59 (56%) were assessed based on overall score to be at high or very high risk of bias (S2 Table). Of the remaining E-O relationships that were not at high/very-high risk of bias, 40 (38%) were assessed as having some concerns, and only six were low risk of bias (6%).

The domain that was most frequently high-risk of bias was Domain 1 (confounding) with over 50% (n = 55, 52%) out of 105 infection-telomere relationships at high risk of bias (Fig 2). Domains 3 (selection) and 4 (post-exposure interventions) also posed challenges, though they were less frequently classified as high risk of bias than Domain 1. However, > 85% of infection-telomere relationships were still rated as having 'some concerns' or higher in each of these domains. Domains 2 (measurement of exposure), 5 (missing data), 6 (measurement of outcome), and 7 (selection of reported result) were mostly rated low risk, with over 80% of E-O relationships classified as low risk.

Table 2. Characteristics of included studies.

| Author & Year | Country & Setting | Population size | Infection/s | Single or multiple TL measurements | Technique of telomere measurement e.g. qPCR | Type of telomere length measure, e.g., relative or absolute | Cell type | Study design |
|---|---|---|---|---|---|---|---|---|
| Aiello (2017) | USA, Six US communities | 163 | C. pneumoniae, Cmv, hsv-1, h.pylori, Combined burden | Single | Q-PCR | Relative, T/S ratio | Leukocyte | Cross-sectional |
| Al-Awadhi (2024) | Kuwait, Mubarak Al-kabeer Hospital | 287 | high risk hpv | Single | Q-PCR | Relative, T/S ratio | Cervical epithelial cells | Cross-sectional |
| Albosale (2021) | Unclear | 100 | HPV | Single | Q-PCR | Relative, T/S ratio | Cervical epithelial cells | Cross-sectional |
| Andreu-Sanchez (2024) | Netherlands, Lifelines cohort | 1,243 | CMV, Rhinovirus | Single | FLOW-FISH | Absolute TL | Lymphocytes, Granulocytes, Naive T-cells, Memory T-cells, B-cells, NK-cells | Cross-sectional |
| Auld (2016) | Uganda, Mulago Hospital in Kampala | 184 | HIV, TB | Single | Q-PCR | Relative, T/S ratio | PBMCs | Cross-sectional |
| Babu (2019) | India, Chennai | 96 | HIV | Single | Q-PCR | unclear | PBMCs | Cross-sectional |
| Benetos (2021) | France, Geriatric Department of the University of Nancy | 38 | COVID-19 | Single | TeSLA and southern blot measurements | absolute | Leukocyte | Cross-sectional |
| Breen (2022) | USA, The Multicenter AIDS Cohort Study (MACS) | 204 | HIV | Multiple | DNA methylation | methylation-based estimate | PBMCs | Cohort |
| Cadinanos (2024) | Spain, La Paz University Hospital,Madrid | 384 | HIV | Single | Q-PCR | Relative, T/S ratio | Blood telomere length | Cross-sectional |
| Chico-Sordo (2022) | Spain, IVI-RMA Madrid clinic | 65 | COVID-19 | Single | Q-FISH | Absolute | Granulosa Cells and peripheral blood mononuclear cells | Cross-sectional analysis within a cohort study |
| Ding (2018) | China, Taizhou prefecture of Zhejiang province | 488 | HIV | Single | Q-PCR | Relative, T/S ratio | Leukocyte | Cross-sectional |
| Dowd (2013) | UK, Data are drawn from the Whitehall II study | 434 | CMV | Single | Q-PCR | Relative, T/S ratio | PBMC | Cross-sectional |
| Dowd (2017) | UK, Participants were from the Heart Scan subsample of the Whitehall II epidemiological cohort (consists of British civil servants from 20 departments) | 400 | Cmv, hsv-1, hhv-6, ebv, Combined burden | Multiple | Q-PCR | Relative, T/S ratio | Leukocyte | Cross-sectional and cohort |
| Freimane (2021) | Latvia, | 108 | multi drug resistant TB | Single | Q-PCR | Relative, T/S ratio | Peripheral blood cells | Cross-sectional |
| Gaardbo (2013) | Denmark, Danish hospitals | 98 | HIV | Single | Q-PCR | Relative telomere length compared to uninfected controls | CD8 + enriched PBMC | Cross-sectional |

*(Continued)*

| Author & Year | Country & Setting | Population size | Infection/s | Single or multiple TL measurements | Technique of telomere measurement *e.g. qPCR* | Type of telomere length measure, *e.g., relative or absolute* | Cell type | Study design |
|---|---|---|---|---|---|---|---|---|
| Giesbrecht (2014) | Canada, British Colombia | 126 | HIV | Single | Q-PCR | Relative, T/S ratio | Peripheral blood cells | Cross-sectional |
| Gogia (2015) | USA, San Francisco General Hospital | 89 | HIV | Single | Q-PCR | Relative, T/S ratio | PBMCs | Cross-sectional |
| Gonzalez-Serna (2017) | Canada, A subset of the Vancouver Injection Drug User Study | 95, unclear follow-up | HIV, HCV | Multiple | Q-PCR | Relative, T/S ratio | PBMCs | Retrospective cohort |
| Grady (2013) | Netherlands, Amsterdam | 74 | HCV mono-infection, HCV-HIV coinfection | (Change) Two Measurements, Not Clear Which Time Points | Flowcytometry and fluorescent in situ hybridization (FLOW-FISH) | Median telomere length of T cell subsets relative to telomere length of calf thymocytes | T cells | Cohort |
| Hampras (2016) | USA, University of South Florida and Moffitt Cancer Center | 336 | HPV | Single | Q-PCR | Relative, T/S ratio | Peripheral blood leukocytes | Cross-sectional analysis in case-control study |
| Hartling (2013) | Denmark, Copenhagen | 75 | HCV | Single | Q-PCR | Relative, T/S ratio | PBMCs | Cross-sectional |
| Hsieh (2015) | Unclear | 'n' differs by cell type with maximum 'n' of 29 | HIV | Single | Q-PCR | Relative, T/S ratio | PBMCs including: Proliferative CD8+CD28+T cells, senescent CD8+CD28- T cells, CD4+ | Cross-sectional analysis in cohort study |
| Huang (2020) | USA | 3,472 | H.pylori | Single | Q-PCR | Relative, T/S ratio | Leukocyte | Cross-sectional |
| Huang (2022) | European cohorts | 1,388,342 critically ill COVID-19. 472,174 LTL | Critically-ill COVID-19 | Single | Q-PCR | Relative, T/S ratio | Leukocyte | Mendelian randomization study |
| Imam (2012) | Canada, British Columbia Women's Hospital, Vancouver, Canada | 99 | HIV | Single | Q-PCR | Relative, T/S ratio | Leukocyte | The relevant data was cross-sectional data but collected from a prospective cohort study |
| Jiang (2022) | European cohorts | LTL: 78,592, Covid-19 susceptibility: 1,683,768 Covid-19 severity: 1,887,658 | COVID-19 | Single | Q-PCR | Relative, T/S ratio | Leukocyte | Two-sample bidirectional Mendelian Randomization Study |
| Jiang (2023) | UK, UK Biobank | telomere length =472,174. Sepsis = 486,484. | Sepsis | Single | Q-PCR | Relative, T/S ratio | Leukocyte | Bidirectional Mendelian randomization (MR) study |
| Jimenez (2016) | Netherlands, Amsterdam | 189 | HIV | Single | Q-PCR | Relative, T/S ratio | PBMCs | Cross-sectional analysis within cohort study |

*(Continued)*

**Table 2.** (Continued)

| Author & Year | Country & Setting | Population size | Infection/s | Single or multiple TL measurements | Technique of telomere measurement e.g. qPCR | Type of telomere length measure, *e.g., relative or absolute* | Cell type | Study design |
|---|---|---|---|---|---|---|---|---|
| Krasnienkov (2022) | Ukraine, Kyiv | 106 | COVID-19 | Single | Q-PCR | Relative, T/S ratio | Peripheral blood leukocytes | Cross-sectional |
| Liang (2024) | USA, Veterans Aging Cohort Study and Women's Interagency HIV Study Cohort | Veterans Aging Cohort Study = 1251 Women's Interagency HIV Study Cohort = 481 | HIV | Single | Methylation | Relative, T/S ratio | PBMCS + whole blood | Cross-sectional |
| Liu (2015) | Canada, Canada (several locations) | 922 | HIV | Single | Q-PCR | absolute TL | Peripheral blood leukocytes | Cross-sectional |
| Ma (2016) | China, hospital of North Sichuan Medical College, Nanchong, Sichuan | 396 | HBV | Single | Q-PCR | Relative, T/S ratio | Peripheral blood leukocytes | Cross-sectional |
| Macamo (2024) | South Africa, primary healthcare facilities | 100 | HIV, Helminths, HIV + Helminths | Single | Q-PCR | Relative, T/S ratio | Leukocyte | Cross-sectional |
| Malan (2015) | South Africa, Dr Kenneth Kaunda Education district in the North West Province | 341 | HIV | Single | Q-PCR | Relative, T/S ratio | Leukocyte | Cross-sectional |
| Malan-Muller (2013) | South Africa | 128 | HIV | Single | Q-PCR | Relative, T/S ratio | Leukocyte | Cross-sectional |
| Manavalan (2016) | USA, Columbia University Medical Center (CUMC) in New York City | 45 | HIV | Single | Q-PCR | Relative, T/S ratio | Peripheral blood osteogenic precursor (COP) cells | Cross-sectional |
| Mehta (2021) | USA, Translational Methamphetamine AIDS Research Center, San Diego | 161 | HIV | Single | Q-PCR | Relative, T/S ratio | Leukocyte | Cross-sectional |
| Meijers (2013) | Unclear | 159 | CMV | Single | Flow-FISH | Relative, T/S ratio | Peripheral blood mononuclear cells (CD4+ and CD8 + T cells) | Cross-sectional |
| Miedema (1996) | Unclear, Unclear | 21, (unclear follow-up) | HIV | Multiple | Unclear | Absolute TRF length | PBMCS | Cohort |
| Mongelli (2021) | Unclear, Unclear | 261 | COVID-19 | Single | Q-PCR | Absolute TL | Un-specified peripheral blood cells | Cross-sectional |
| Muhsen (2019) | Israel, Jerusalem | 934 | H.Pylori | Single | Southern blot | Absolute TL | Leukocyte | Cross-sectional |
| Nguyen (2022) | USA | 3454 | Periodontitis | Single | Q-PCR | Absolute (bp) and relative TL (T/S ratio) | Leukocyte | Cross-sectional |

*(Continued)*

**Table 2.** (Continued)

| Author & Year | Country & Setting | Population size | Infection/s | Single or multiple TL measurements | Technique of telomere measurement e.g. qPCR | Type of telomere length measure, e.g., relative or absolute | Cell type | Study design |
|---|---|---|---|---|---|---|---|---|
| Noppert (2020) | USA, Data from a large, nationally representative sample of US adults (National Health and Nutrition Examination Survey (NHANES) | 1708 | HSV-1, HSV-2, CMV, H.pylori, HBV, Combined burden | Single | Q-PCR | Relative, T/S ratio | Leukocyte | Cross-sectional study |
| Panczyszyn (2020) | Unclear | 88 | high-risk hpv | Single | Q-PCR | absolute | Non-specific blood and cervical cells | Cross-sectional |
| Pathai (2013) | South Africa, Clinics in township communities in Cape Town | 486 | HIV | Single | Q-PCR | Relative, T/S ratio | Leukocyte | Cross-sectional |
| Petrara (2024) | Italy, Department of Women's and Children's Health of the University Hospital of Padova | 78 | HIV | Single | Q-PCR | Relative, T/S ratio | PBMCs | Cross-sectional |
| Retuerto (2022) | Spain, Madrid | 420 | COVID-19 | Single | Q-PCR | Absolute TL | Peripheral blood leukocytes | Cross-sectional |
| Richardson (2000) | USA and France | 170 | HIV | Single | Southern Blot | TRF length | PBMCs | Cross-sectional |
| Saberi (2019) | Canada, Vancouver | 105, unclear follow-up time | HIV, HCV | Multiple | Q-PCR | Relative TL (T/S ratio) | Leukocyte | Cohort |
| Savrun (2024) | Turkey,Emergency Department at Ordu University | 140 | COVID-19 | Single | Q-PCR | Relative, T/S ratio | Un-specified blood cells | Cross-sectional |
| Sehl (2021) | USA, Baltimore, Pittsburgh, Los Angeles, Chicago | 201 | HIV, HBV | Multiple | Methylation | DNAm-TL | Peripheral blood mononuclear cells | Cohort |
| Shiau (2021) | USA, Columbia University Irving Medical Center (CUIMC) in New York City | 107 | HIV | Single | Methylation | DNAm-TL | Not clear- says blood extracted | Cross-sectional |
| Shiau (2024) | USA, 3 Women's Interagency HIV Study sites (San Francisco, Bronx, and Chicago) | 190 | HIV | Single | Methylation | DNAmTL | PBMCs | Cross-sectional |
| Soares (2025) | Brazil, São Paulo | 112 | COVID-19 | Single | Q-PCR | Relative, T/S ratio | Sperm cells | Cross-sectional |
| Song (2020) | USA, The United States | 3,478 | periodontitis | Single | Q-PCR | Relative (T/S ratio) and absolute TL (base pairs) | Leukocyte | Cross-sectional |
| Spyridopoulos (2009) | Unclear | 33 | CMV | Single | Flow-FISH | absolute | Leukocyte (various types) | Cross-sectional |
| Srinivasa (2014) | USA, Boston | 142 | HIV | Single | Q-PCR | Relative, T/S ratio | Leukocyte | Cross-sectional |
| Tachtatzis (2011) | unclear, unclear | 94 | HBV | Single | Q-FISH | Mean fluorescent intensity | Hepatocytes | Cross-sectional |

*(Continued)*

| Author & Year | Country & Setting | Population size | Infection/s | Single or multiple TL measurements | Technique of telomere measurement e.g. qPCR | Type of telomere length measure, e.g., relative or absolute | Cell type | Study design |
|---|---|---|---|---|---|---|---|---|
| Tahara (2013) | Unclear | 150 | h.pylori | Single | Q-PCR | Relative, T/S ratio | Gastric mucosa cells | Cross-sectional |
| Toljic (2023) | Serbia, Clinic for Infectious and Tropical Diseases, School of Medicine, University of Belgrade and unexposed blood donor volunteers from the Blood Transfusion Institute of Serbia, Belgrade | 205 | HIV | Single | Q-PCR | Relative, T/S ratio | Leukocytes of whole blood | Cross-sectional |
| Tucker (2000) | UK, West London hospitals | 21 | HIV | Single | Southern blot | TRF length measured to calculate absolute TL | Lymphocytes (4 cell types) | Cross-sectional |
| Usadi (2016) | USA, 5 major US blood centers (Baltimore/ Washington, Detroit, Oklahoma City, San Francisco, and Los Angeles) | 135 | HTLV-1, HTLV-2 | Single | Q-PCR | Relative, T/S ratio | Peripheral blood mononuclear cells | Cross-sectional |
| Wang (2019) | Uganda, Mulago National Referral Hospital in Kampala, | 434 | HIV, TB | Single | Q-PCR | Relative, T/S ratio | PBMCs | Cross-sectional |
| Wang (2022) | China, Shandong University Second Hospital | 1318 | high risk hpv | Single | Q-PCR | Relative, T/S ratio | Epithelial cells | Cross-sectional |
| Womersley (2021) | South Africa, community health care facilities in and around Cape Town | 286 (cross-sectional), 110 (longitudinal) | HIV | Multiple | Q-PCR | Relative, T/S ratio | Not clear, states that DNA was extracted from whole blood | Cross-sectional and cohort |
| Woods (2023) | USA, San Diego County | 149 | HIV | Single | Q-PCR | Relative, T/S ratio | not clear | Cross-sectional |
| Xu (2022) | European cohorts, European ancestry studies | telomere length: 472174. Covid susceptibility: (1,683,768). Covid severity: (1,388,342) | COVID-19 | Single | Q-PCR | Relative, T/S ratio | Leukocyte | Bidirectional Mendelian randomization study |
| Xu (2024) | UK, UK Biobank | telomere length =472,174. Sepsis = 486,484. | Sepsis | Single | Q-PCR | Relative, T/S ratio | Leukocyte | Bidirectional Mendelian randomization (MR) study |
| Yang (2024) | Canada, Canada (several locations) | 376 | HCV, HHV-8, HSV-2, CMV, HSV-1, EBV, HIV, combined burden | Single | Q-PCR | Relative, T/S ratio | Leukocyte | Cross-sectional |

*(Continued)*

**Table 2.** (Continued)

| Author & Year | Country & Setting | Population size | Infection/s | Single or multiple TL measurements | Technique of telomere measurement e.g. qPCR | Type of telomere length measure, e.g., relative or absolute | Cell type | Study design |
|---|---|---|---|---|---|---|---|---|
| Yoshioka (2012) | Japan, the Endoscopy Center of Fujita Health University Hospital | 150 | H. Pylori | Single | Q-PCR | Relative, T/S ratio | Leukocyte | Cross-sectional |
| Zanet (2014) | Canada, Vancouver | 395 | HIV, HCV, HBV | Single | Q-PCR | Relative, T/S ratio | Leukocyte | Cross-sectional |
| Zhang (2014) | China, Department of Thoracic Surgery in the Affiliated Tumor Hospital of Shantou University Medical College | 70 | high-risk hpv | Single | Q-PCR | Relative, T/S ratio | Oesophageal squamous cell carcinoma (ESCC) and paired matched adjacent noncancerous tissues | Cross-sectional |
| Zribi (2019) | Israel, 14-bed, general intensive care department of the Rabin Medical Center, Campus Beilinson, Israel | 40 | Sepsis | 2 Measures | Q-PCR | Relative, T/S ratio | Unspecified blood cells | Cohort |

Abbreviations: TL = Telomere length, Q-PCR = Quantitative Polymerase Chain Reaction, TRF = Terminal Restriction Fragment, PBMCs = Peripheral Blood Mononuclear Cells, SNP = Single Nucleotide Polymorphism, MR = Mendelian Randomization, ART = Antiretroviral Therapy, CMV = Cytomegalovirus, HCV = Hepatitis C Virus, HBV = Hepatitis B Virus, HPV = Human Papillomavirus, HTLV = Human T-Lymphotropic Virus, NHANES = National Health and Nutrition Examination Survey, ELISA = Enzyme-Linked Immunosorbent Assay, FISH = Fluorescence In Situ Hybridization, T/S Ratio = Telomere-to-Single Copy Gene Ratio, DNAmTL = DNA methylation-based telomere length.

## Infection type

Across the 73 studies, 22 separate infections and 6 different co-infections were investigated. The most frequent pathogen was HIV, found in 35 of the 73 studies (48%), followed by Coronavirus Disease 2019 (COVID-19) (10 studies), Cytomegalovirus (CMV) (8 studies), Helicobacter Pylori (H.Pylori) and hepatitis C with 6 studies each and hepatitis B (5 studies). All other infections had 3 or fewer studies.

The 105 E-O relationships examined in the 73 studies are displayed stratified by infection type, cell type, TL or change in TL, TL measure, TL measurement assay and statistical analysis method in S3 Table. There were a total of 89 unique E-O relationships. Very few unique E-O relationships were homogeneous with respect to infection type, cell type, outcome (TL or change in TL), TL measure (relative or absolute), TL measurement assay, and statistical method. Only HIV, Herpes Simplex Virus-1 (HSV-1), CMV and H. pylori had more than two unique homogeneous E-O relationships.

## The association of infections with TL

Overall, across all infections, 42 of 89 (47%) unique exposure–outcome relationships showed evidence of association (p < 0.05), with infection linked to shorter TL or greater telomere attrition (Table 3). However some of these results were unadjusted, of the unique E-O relationships with matching or adjusted analysis 23 of 55 (42%) showed evidence of association (p < 0.05).

By infection type, 23 out of 29 HIV unique E-O relationships (79%) showed an association, and of the unique HIV E-O relationships with matching or adjusted analysis, 15 out of 19 (79%) showed an association. For other pathogens,

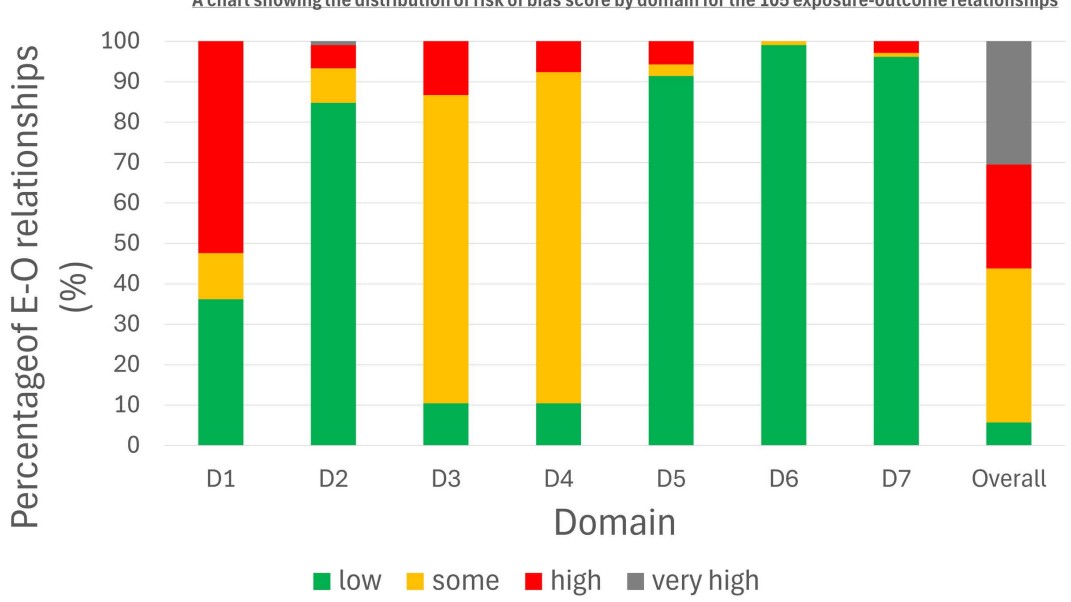

**A chart showing the distribution of risk of bias score by domain for the 105 exposure-outcome relationships**

**Fig 2. Risk of Bias by Domain Across 105 Exposure–Outcome Relationships. D1-confounding, D2-measurement of the exposure, D3- selection of participants into the study (or into the analysis), D4-post-exposure interventions, D5- missing data, D6- measurement of the outcome, D7- selection of the reported result.**

evidence was variable and the number of E-O relationships for other pathogens were small, often with only one unique E-O relationship available for each infection type. Four studies investigated the combined burden of multiple infections, each looking at different combinations of pathogens; one showed an association between infection burden and TL/TA, whereas the other three did not.

Depicted in Forest plots are difference in mean TL by infection status for HIV *(Fig 3)* (N=11), CMV *(Fig 4)* (N=4), and HSV-1 *(Fig 5)* (N=4). Most HIV studies presenting difference in mean TL showed an association between HIV and reduced TL or increased telomere attrition. For CMV and HSV-1 the results were mixed, with only one of three studies (all three studies presented results for both pathogens) showing an association between infection and reduced TL for both pathogens.

## Infection severity

The relationship between infection severity and TL was examined in studies of 34 E-O relationships, yielding mixed results (S3 Table). There were insufficient homogenous studies with respect to severity, which made comparisons difficult. The most commonly used severity measures for HIV were viral load (n=4) and HIV progression (fast vs no/slow progression, n=4). For viral load, evidence of an association between increased viral load and shorter telomere length was found in two [38,79] of four E-O relationships. For progression status, faster progression was associated with shorter telomere length in three [36,60,74] out of four E-O relationships where this was evaluated.

For other pathogens such as CMV and COVID-19 results were less clear and conflicting.

## Age and sex

Relatively few studies investigated whether the association between infection and telomere length differed according to age or sex. None found strong evidence for interaction, either due to p-values greater than 0.05 or because no statistical test was performed (S3 Table).

**Table 3. Exposure-outcome relationships by pathogen/infection type.**

| Pathogen/infection | Number of E-O relationships examined | Number of unique E-O relationships[1] | Maximum number of unique homogenous E-O relationships[2] | Number of unique E-O relationships showing evidence infection was associated (p < 0.05) with shorter telomere length or greater attrition (% of all unique E-O relationships) [3] | Number of unique E-O relationships with any matching/ adjusted analysis | Number of unique E-O relationships with matching/ adjusted analysis AND showing evidence infection was associated (p < 0.05) with shorter telomere length or greater attrition (% of all unique E-O relationships with matching/ adjusted analysis) | Number of unique E-O relationships with high/ very high overall risk of bias score (% of all unique E-O relationships) |
|---|---|---|---|---|---|---|---|
| HIV | 35 | 29 | 4 | 23 (79.3) | 19 | 15 (78.9) | 21 (72.4) |
| COVID-19 | 10 | 8 | None | 5 (62.5) | 3 | 2 (66.7) | 7 (87.5) |
| CMV | 8 | 7 | 4 | 2 (28.57) | 5 | 1 (20.0) | 3 (42.9) |
| H. Pylori | 6 | 4 | 2 | 1 (25.0) | 3 | 0 | 1 (25.0) |
| HCV | 6 | 4 | None | 1 (25.0) | 2 | 1 (50.0) | 4 (100.0) |
| HBV | 5 | 5 | 2 | 2 (40.0) | 1 | 0 | 4 (80.0) |
| HPV | 2 | 2 | None | 1 (50.0) | 0 | 0 | 2 (100.0) |
| HR-HPV | 4 | 4 | None | 1 (25.0) | 0 | 0 | 4 (100.0) |
| HSV-1 | 4 | 4 | 4 | 1 (25.0) | 4 | 1 (25.0) | 0 |
| Sepsis | 3 | 2 | None | 0 | 1 | 0 | 1 (50.0) |
| EBV | 2 | 2 | 2 | 0 | 2 | 0 | 0 |
| HSV-2 | 2 | 2 | 2 | 0 | 2 | 0 | 0 |
| TB | 2 | 1 | None | 0 | 0 | 0 | 1 (100.0) |
| MDR-TB | 1 | 1 | None | 1 (100.0) | 0 | 0 | 1 (100.0) |
| Periodontitis | 2 | 1 | None | 0 | 1 | 0 | 0 |
| C. pneumoniae | 1 | 1 | None | 0 | 1 | 0 | 0 |
| Helminths | 1 | 1 | None | 1 (100.0) | 1 | 1 (100.0) | 0 |
| HHV-6 | 1 | 1 | None | 0 | 1 | 0 | 0 |
| HHV-8 | 1 | 1 | None | 0 | 1 | 0 | 0 |
| HTLV-1 | 1 | 1 | None | 0 | 1 | 0 | 1 (100.0) |
| HTLV-2 | 1 | 1 | None | 0 | 1 | 0 | 1 (100.0) |
| Rhinovirus | 1 | 1 | None | 1 (100.0) | 0 | 0 | 1 (100.0) |
| HIV/HCV co-infection | 1 | 1 | None | 1 (100.0) | 1 | 1 (100.0) | 1 (100.0) |
| HIV + Helminths | 1 | 1 | None | 0 | 1 | 0 | 0 |
| Combined Burden of HSV-1,HSV-2, CMV, H. Pylori, HBV | 1 | 1 | None | 0 | 1 | 0 | 0 |
| Combined burden of C. pneumoniae, HSV-1, CMV, H. Pylori | 1 | 1 | None | 0 | 1 | 0 | 0 |
| Combined burden of CMV, HSV-1, HHV-6, EBV | 1 | 1 | None | 1 (100.00) | 1 | 1 (100.0) | 0 |

*(Continued)*

**Table 3.** (Continued)

| Pathogen/ infection | Number of E-O relationships examined | Number of unique E-O relationships[1] | Maximum number of unique homogeneous E-O relationships[2] | Number of unique E-O relationships showing evidence infection was associated (p<0.05) with shorter telomere length or greater attrition (% of all unique E-O relationships)[3] | Number of unique E-O relationships with any matching/ adjusted analysis | Number of unique E-O relationships with matching/ adjusted analysis AND showing evidence infection was associated (p<0.05) with shorter telomere length or greater attrition (% of all unique E-O relationships with matching/ adjusted analysis) | Number of unique E-O relationships with high/ very high overall risk of bias score (% of all unique E-O relationships) |
|---|---|---|---|---|---|---|---|
| Combined burden of CMV, EBV, HCV, HHV-8, HIV, HSV-1, HSV-2 | 1 | 1 | None | 0 | 1 | 0 | 0 |
| **Total** | 105 | 89 | N/A | 42 (47.2) | 55 | 23 (41.8) | 53 (60.0) |

1- E-O relationships were considered 'unique' if they derived their results from distinct samples (i.e., no overlapping data). When multiple E-O relationships for the same infection reported on the same population, we selected a single study to represent that population, prioritising longitudinal designs, then those with the largest sample sizes, and finally the most recent publications.

2- This column shows the largest number of homogeneous E-O relationships for each specific pathogen or infection. E-O relationships were deemed 'homogeneous' if they were directly comparable to each other, i.e., they matched across all the potential sources of heterogeneity (telomere length (TL) or change in TL, TL measure, TL measurement assay, and statistical method). For example, if among the CMV studies there were two sets of comparable E-O relationships; one set consisting of two E-O relationships, and the other of three, the largest group of comparable studies would be three. Hence "3" would be presented in the table.

3- Some studies had mixed results. This column indicates if their primary outcome was associated with reduced TL

Abbreviations: HIV: Human Immunodeficiency Virus, COVID-19: Coronavirus Disease 2019, CMV: Cytomegalovirus, H. pylori: Helicobacter pylori, HCV: Hepatitis C Virus, HBV: Hepatitis B Virus, HPV: Human Papillomavirus, HR-HPV: High-Risk Human Papillomavirus, HSV-1: Herpes Simplex Virus Type 1, TB: Tuberculosis, MDR-TB: Multidrug-Resistant Tuberculosis, C. pneumoniae: Chlamydia pneumoniae, EBV: Epstein-Barr Virus, E-O: Exposure-outcome, HHV-6: Human Herpesvirus 6, HSV-2: Herpes Simplex Virus Type 2, HTLV-1: Human T-Lymphotropic Virus Type 1, HTLV-2: Human T-Lymphotropic Virus Type 2

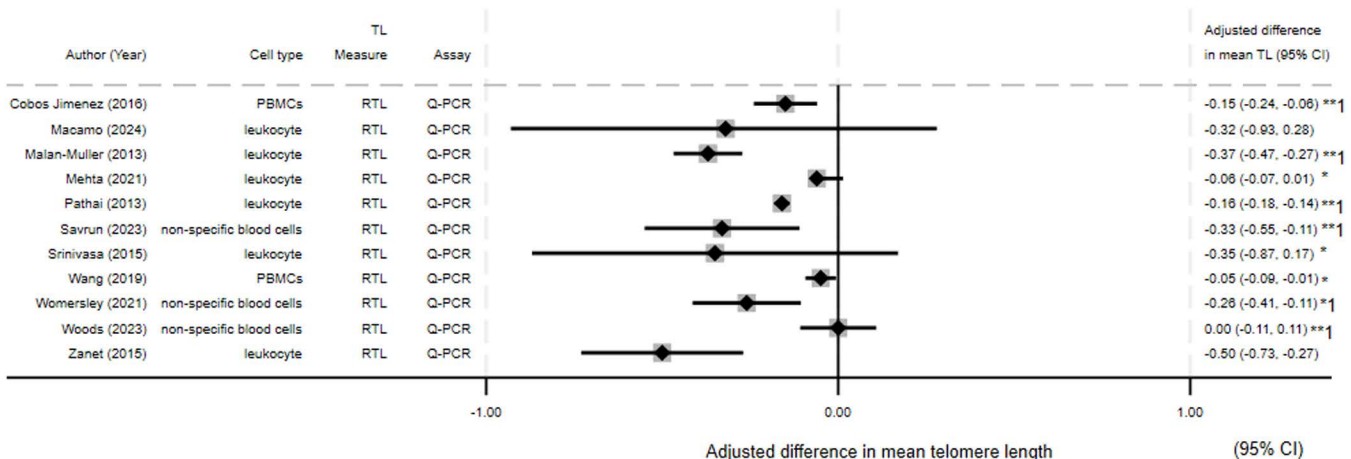

**Fig 3. Forest plot of HIV studies presenting difference in mean telomere length.** The difference in means relates to mean in infected group minus mean in control group. 1- Results are from univariate analysis only. *−95% Confidence interval calculated from available data. **- Effect estimate and 95% confidence interval calculated from available data.

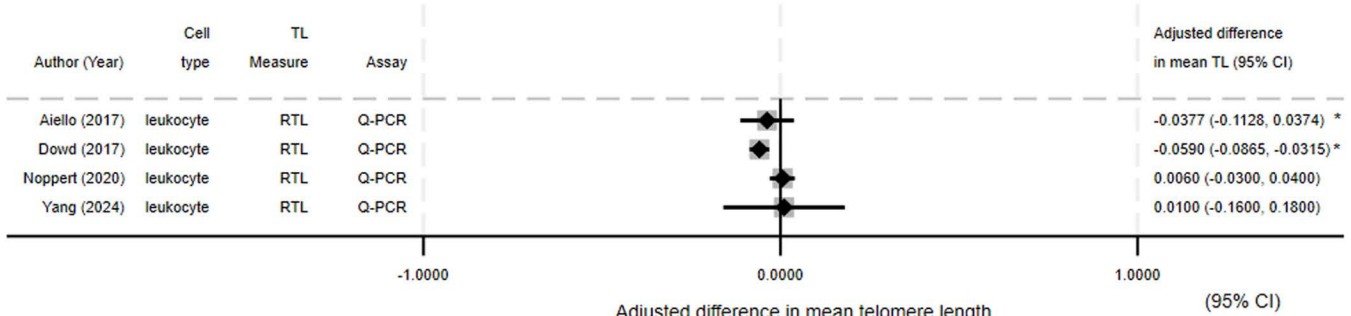

**Fig 4. Forest plot of CMV studies presenting difference in mean telomere length.** The difference in means relates to mean in infected group minus mean in control group. *−95% Confidence interval calculated from available data.

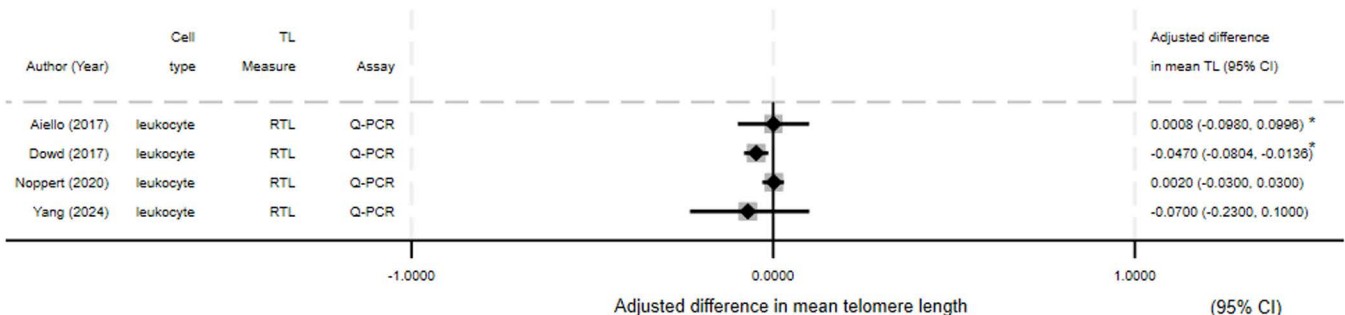

**Fig 5. Forest plot of HSV-1 studies presenting difference in mean telomere length.** The difference in means relates to mean in infected group minus mean in control group. *−95% Confidence interval calculated from available data.

## GRADE assessment

The overall evidence on the associations of HIV, CMV and HSV-1 difference in mean telomere length were classified as of very low quality using the GRADE assessment tool (Table 4). For HIV this was due to the observational nature of the studies as well as issues with inconsistency, imprecision, and indirectness. For HSV-1 and CMV, all studies were observational and there was issues with inconsistency.

For the publication bias domain, funnel plots were not performed for any of the infection types. Although the HIV exposure group met the conventional minimum threshold for funnel plot analysis (n = 10 studies) [95], 6 of these reported

Table 4. GRADE quality assessment.

| Outcome | Expo-sure | Study design and no. of studies | Risk of bias | Incon-sistency | Indirect-ness | Impre-cision | Publica-tion bias | Upgrade | Quality |
|---|---|---|---|---|---|---|---|---|---|
| **Difference in mean blood cell telomere length** | **HIV** | 11 cross-sectional studies | Very serious | Very serious | Very serious | Not serious | N/A | None | ⊕○○○ Very low |
| | **CMV** | 4 cross-sectional studies | Not serious | Serious | Not serious | Not serious | N/A | None | ⊕○○○ Very low |
| | **HSV-1** | 4 cross-sectional studies | Not serious | Serious | Not serious | Not serious | N/A | None | ⊕○○○ Very low |

unadjusted effect estimates. This limited comparability with adjusted estimates and introduced systematic bias unrelated to publication bias. Consequently, an assessment of publication bias was deemed inappropriate in this context.

## Discussion

### Summary of key findings

Our systematic review aimed to summarise the relationship between infections and TL or telomere attrition across various study designs and infection types. We included 73 studies examining 105 E-O relationships between infections and telomere length or telomere attrition. The most frequently represented infection was HIV, which was consistently associated with reduced TL or increased telomere attrition (79% of E-O relationships). In contrast, evidence for other infections was more mixed. Of the four studies investigating pathogen burden, one [15] reported a potential dose-response effect with greater telomere attrition with seropositivity to an increasing number of persistent pathogens, however the other three [17,27,93] reported no association between pathogen burden and TL/ telomere attrition.

Fifty-nine (56%) of the 105 E-O relationships were rated as having a high- or very-high- risk of bias, suggesting that the results may not be reliable. Consequently, while there is some evidence for an association between infections and TL, the overall validity findings may be limited due to potential bias of many included studies.

### Strengths and limitations

Several methodological strengths enhance the reliability of this systematic review. Firstly, there was comprehensive study selection. Our review included a variety of study designs, including cross-sectional, cohort, and Mendelian randomisation studies, allowing for a more comprehensive understanding of the infection-telomere relationship. We included studies from multiple geographical locations and across a range of pathogen types, allowing us to evaluate the potential differential impact of infections across populations. The use of independent reviewers to screen studies and extract data minimised selection bias and allowed for reproducible findings. Our approach of categorising studies based on infection type, cell type, telomere measurement approach, and statistical analysis method to assess homogeneity and uniqueness allowed for meaningful comparisons. Finally, the assessment of risk of bias for each exposure-outcome relationship as well as GRADE assessment for comparable HIV, HSV-1 and CMV studies, provides information on the quality and reliability of the studies included.

However, our systematic review had several limitations potentially affecting the interpretation of its findings. We were unable to systematically identify grey literature due to a cyber-attack on the British library meaning a planned search of the EThOS database was not possible [96]. The second reviewer only conducted 10% random sample of data extraction so some of the extracted data has not been verified, although this is unlikely to have affected results.

In terms of the limitations of the included literature, we saw considerable heterogeneity with respect to infection type, cell type, telomere measurement approaches and statistical analysis method. This between-study heterogeneity limited comparison across studies meaning a meta-analysis was not possible as a pooled analysis would have been difficult to interpret.

The high or very high risk of bias found in most studies, especially concerning confounding (ROBINS-E Domain 1) and participant selection (Domain 2), means the results may not be reliable estimates of the effect of infection on TL. Many included studies had inadequate control for confounders such as age, sex, ethnicity, and comorbidities. This inadequate accounting for confounding means that many of the associations could be biased as estimates of the effect of infection on TL. Additionally, issues with participant selection such as unclear recruitment strategies or the inclusion of participants already infected at enrolment, raise concerns about selection bias and further weaken the evidence. These methodological shortcomings highlight the need for more rigorously designed studies to reduce bias and improve the validity of findings. Most included studies were cross-sectional, which limits the ability to establish causality between infection and TL. Longitudinal studies are essential for determining whether infections lead to telomere shortening or if individuals with

shorter telomeres are more susceptible to certain infections. Previous studies have suggested the latter, with shorter TL associated with greater susceptibility to viral infection and worse clinical outcomes [97,98], highlighting the possibility of a reciprocal relationship between infection and TL akin to the "chicken or egg" dilemma.

Most studies focused on chronic infections, with only a few examining the effect of acute infections like COVID-19 on TL. Understanding the effect of acute versus chronic infections on telomere dynamics may provide a more comprehensive picture of infection-related telomere attrition. Moreover, we found no studies examining vaccination or infection treatments as exposures which limits our ability to assess the likely effectiveness of anti-infective interventions to prevent telomere shortening [22].

## Comparison with related literature

To our knowledge, there are no previous systematic reviews of this topic. A previous study [99] (that did not meet our inclusion criteria due to a lack of appropriate control group) found that a one unit increase of CMV antibody IgG titre was associated with −0.06 (95% confidence interval: −0.11, −0.01) unit decrease of leukocyte TL after adjusting for age, sex, body mass index and smoking status. These findings are consistent with just one [15] of the unique CMV studies in the present review, with the other four showing no association or inconclusive evidence.

Another study (again excluded due to lack of uninfected comparators) found that the shortest TL was observed three months post malaria infection compared to day 0, but that TL recovered after 12 months post-infection when there was no longer evidence of difference in TL [100]. This highlights the mixed results observed in our systematic review, suggesting that the measurement of TL and the time since infection may require further investigation.

Our review was restricted to studies within adults, however other individual studies [101,102] compared the effect of HIV infection vs non-infection on TL in children. The result of these studies in children were mixed, with one [101] concluding that absolute TL was shorter in HIV-infected and HIV-exposed uninfected (HEU) children compared with HIV-unexposed uninfected (HUU) children, but did not differ between HIV-infected and HEU children. Another study involving children found no associations between children's LTL and perinatal antiretroviral therapy (ART) exposure or HIV status. This highlights the mixed results observed in our systematic review, suggesting that the measurement of TL and the time since infection may require further investigation [102]. However, they did find an association between having a detectable HIV viral load and shorter LTL. The authors suggested that these results showed that uncontrolled HIV viremia may be associated with acceleration of telomere attrition.

## Implications for future research

Future research could focus on under-studied acute infections, employ longitudinal study designs to establish temporality, use standardised telomere measurement methods to increase inter-study comparability, and apply comprehensive adjustment for confounders. Immune ageing, measured via telomere length as well as other methods such as epigenetic clocks [103], could be explored as a mechanism explaining the relationship between infections and age-related diseases.

## Conclusions

Our systematic review highlights a potential association between infection and accelerated immune ageing, measured by TL and attrition, particularly in HIV. However, the evidence is limited by methodological issues and rated very low quality overall. Addressing these limitations through more robust longitudinal designs, standardised measurement methods, and a focus on adjustment for confounding factors will improve the quality of studies addressing this relationship.

## Supporting information

**S1 File. PRISMA 2020 checklist.**
(DOCX)

**S2 File. Infection and telomere length: a systematic review protocol.**
(PDF)

**S3 File. Search strategy.**
(DOCX)

**S4 File. GRADE quality assessment reasons to up- or downgrade.**
(DOCX)

**S1 Table. Data extraction form details.**
(DOCX)

**S2 Table. Overall and domain-specific risk of bias scores for each exposure-outcome relationship using the ROBINS-E tool.**
(DOCX)

**S3 Table. Table of 105 exposure-outcome relationships grouped by infection type, cell type, whether the outcome was TL or change in TL, TL measure (relative or absolute), TL measurement assay, and statistical method.**
(DOCX)

## Acknowledgments

We would like to acknowledge Russel Burke, Assistant Librarian at the London School of Hygiene and Tropical Medicine. His invaluable support helped develop the search strategy for this systematic review.

## Author contributions

**Conceptualization:** Louis Tunnicliffe, Charlotte Warren-Gash.

**Data curation:** Louis Tunnicliffe.

**Formal analysis:** Louis Tunnicliffe, Jonathan W Bartlett.

**Funding acquisition:** Charlotte Warren-Gash.

**Investigation:** Louis Tunnicliffe.

**Methodology:** Louis Tunnicliffe, Rutendo Muzambi, Jonathan W Bartlett, Khalid A Basit, Veryan Codd, Charlotte Warren-Gash.

**Supervision:** Rutendo Muzambi, Jonathan W Bartlett, Charlotte Warren-Gash.

**Validation:** Khalid A Basit, Kwabena Asare, Georgia Gore-Langton, Kathryn E Mansfield.

**Writing – original draft:** Louis Tunnicliffe.

**Writing – review & editing:** Louis Tunnicliffe, Rutendo Muzambi, Jonathan W Bartlett, Laura D Howe, Khalid A Basit, Kwabena Asare, Georgia Gore-Langton, Kathryn E Mansfield, Veryan Codd, Charlotte Warren-Gash.

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
