## [Decision Letter · Decision Letter 0]

22 Aug 2025

PONE-D-25-38272Infection and telomere length: a systematic reviewPLOS ONE

Dear Dr. Tunnicliffe,

Thank you for submitting your manuscript to PLOS ONE. After careful consideration, we feel that it has merit but does not fully meet PLOS ONE’s publication criteria as it currently stands. Therefore, we invite you to submit a revised version of the manuscript that addresses the points raised during the review process.

**ACADEMIC EDITOR: Please address the comments of the reviewer carefully.**

We look forward to receiving your revised manuscript.

Kind regards,

Gabriele Saretzki, PhD

Academic Editor

PLOS ONE

Journal Requirements:

This systematic review was supported by Charlotte Warren-Gash’s Wellcome Career

Development Award [225868/Z/22/Z]

5. Please remove all personal information, ensure that the data shared are in accordance with participant consent, and re-upload a fully anonymized data set.

Additional Editor Comments :

Please address the comments of the reviewer carefully.

Reviewers' comments:

Reviewer's Responses to Questions

**Comments to the Author**

1. Is the manuscript technically sound, and do the data support the conclusions?

Reviewer #1: Yes

2. Has the statistical analysis been performed appropriately and rigorously? 

Reviewer #1: Yes

3. Have the authors made all data underlying the findings in their manuscript fully available?

Reviewer #1: Yes

4. Is the manuscript presented in an intelligible fashion and written in standard English?

Reviewer #1: Yes

5. Review Comments to the Author

Reviewer #1: In the present manuscript, Tunnicliffe et al summarize the relationship between infections and TL from the literature. They observed that HIV infection triggered accelerated telomere shortening in most reports, while results are not very consistent in studies involved in non-HIV infection. The data are interesting.

Major points

It has been shown that individuals bearing shorter TL are more susceptible to infection of different viruses or associated with more adverse outcomes than those with longer TL (JAMA, 2013;309:699; Cell Mol Life Sci, 2022;79:110; EBioMedicine. 2021;70: 103485). Thus, the relationship between infection and TL may be reciprocal, like “the chicken or egg causality dilemma”. The authors are suggested to discuss this issue.

Minor points

The abbreviation “TA” usually indicates telomerase activity, which might make readers confused here (at least for reviewer). In addition, TA is not in the abbreviation list.

6. PLOS authors have the option to publish the peer review history of their article (what does this mean? ). If published, this will include your full peer review and any attached files.

**Do you want your identity to be public for this peer review?** For information about this choice, including consent withdrawal, please see our Privacy Policy .

Reviewer #1: No

---

## [Author Response · Author response to Decision Letter 1]

5 Sep 2025

Response to Reviewers

Manuscript ID: PONE-D-25-38272

Title: Infection and telomere length: a systematic review

Dear Dr. Saretzki,

We would like to thank you and the reviewer for your thoughtful and constructive feedback on our manuscript. We appreciate the opportunity to revise our work and have carefully addressed all points raised by the academic editor, the reviewer, and the journal’s requirements. Below we provide a detailed, point-by-point response.

Reviewer #1 Comments

Major Point 1:

It has been shown that individuals bearing shorter TL are more susceptible to infection of different viruses or associated with more adverse outcomes than those with longer TL (JAMA, 2013;309:699; Cell Mol Life Sci, 2022;79:110; EBioMedicine. 2021;70:103485). Thus, the relationship between infection and TL may be reciprocal, like “the chicken or egg causality dilemma”. The authors are suggested to discuss this issue.

Response:

We thank the reviewer for their valid point, we have addressed this issue by adding the following to the discussion section:

‘Most included studies were cross-sectional , which limits the ability to establish causality between infection and TL. Longitudinal studies are essential for determining whether infections lead to telomere shortening or if individuals with shorter telomeres are more susceptible to certain infections. Previous studies have suggested the latter, with shorter TL associated with greater susceptibility to viral infection and worse clinical outcome (97,98), highlighting the possibility of a reciprocal relationship between infection and TL akin to the “chicken or egg” dilemma.’

Minor Point 1:

The abbreviation “TA” usually indicates telomerase activity, which might make readers confused here (at least for reviewer). In addition, TA is not in the abbreviation list.

Response:

We have now removed the abbreviation “TA” and instead used the unabbreviated ‘telomere attrition’ throughout the manuscript.

Journal Requirements

1. Manuscript style requirements (formatting, file naming).

Response:

We have now amended the naming and formatting of files, figures and tables according to the PLOSONE criteria.

2. Financial disclosure – role of funders.

Response:

3. Separate captions for each figure.

Response:

We now have separate captions for each figure in-line with the PLOSONE criteria.

4. Captions for Supporting Information files and consistent in-text citations.

Response:

We now have captions for supporting information and have consistent in-text citations.

5. Fully anonymised data set, removal of personal information.

Response:

With regard to the request to remove personal information and provide an anonymised dataset, we would like to clarify that our manuscript is a systematic review and does not involve primary data collection or human participants. As such, no participant-level data are included, and the requirement to anonymise datasets is not applicable in this case.

6. Reviewer-suggested references – evaluation and citation if relevant.

Response:

We have reviewed the cited publications. Cohen et al. (JAMA, 2013) and Wang et al. (EBioMedicine, 2021) are directly relevant, as they provide evidence that shorter telomere length is associated with increased susceptibility to viral infection and adverse outcomes. These have now been cited in the Limitations section, where we discuss the possibility of a reciprocal relationship between infection and telomere length.

7. Reference list accuracy, completeness, and handling of retracted papers.

Response:

We have checked the reference list and the status (retracted or not) of cited papers, we confirm that there are no retracted papers and that the reference list is complete and accurate.

Additional Editor Comments

Please address the comments of the reviewer carefully.

Response:

We believe we have carefully addressed all reviewer comments, making changes to the manuscript where needed.

We believe these revisions have strengthened the manuscript, and we are grateful to the editor and reviewer for their guidance. We look forward to your consideration of our revised submission.

Sincerely,

Louis Tunnicliffe

on behalf of all authors

---

## [Editor Report · Decision Letter 1]

10 Sep 2025

Infection and telomere length: a systematic review

PONE-D-25-38272R1

Dear Dr. Tunnicliffe,

We’re pleased to inform you that your manuscript has been judged scientifically suitable for publication and will be formally accepted for publication once it meets all outstanding technical requirements.

Kind regards,

Gabriele Saretzki, PhD

Academic Editor

PLOS ONE

Additional Editor Comments (optional):

Thanks for addressing the reviewer's comments.
---

## [Editor Report · Acceptance letter]

PONE-D-25-38272R1

PLOS ONE

Dear Dr. Tunnicliffe,

I'm pleased to inform you that your manuscript has been deemed suitable for publication in PLOS ONE. Congratulations! Your manuscript is now being handed over to our production team.

Kind regards,

on behalf of

Dr. Gabriele Saretzki

Academic Editor

PLOS ONE